# Research

materials science/environmental chemistry/nanotechnology

soy protein isolate, carboxymethyl cellulose, avermectin, anti-UV, pH sensitivity

**Authors for correspondence:**
Hongjun Zhou
e-mail: hongjunzhou@163.com
Xinhua Zhou
e-mail: cexinhuazhou@163.com

This article has been edited by the Royal Society of Chemistry, including the commissioning, peer review process and editorial aspects up to the point of acceptance.

# Soy protein isolate-carboxymethyl cellulose conjugates with pH sensitivity for sustained avermectin release

Long Chen, Hongjun Zhou, Li Hao, Huayao Chen and Xinhua Zhou

Key Laboratory of Agricultural Green Fine Chemicals of Guangdong Higher Education Institution, School of Chemistry and Chemical Engineering, Zhongkai University of Agriculture and Engineering, Guangzhou, People's Republic of China

LC, 0000-0003-3336-2115; HZ, 0000-0002-4210-6327

Carboxymethyl cellulose (CMC) was grafted onto the surface of soy protein isolate (SPI) to obtain soy protein isolate-carboxymethyl cellulose conjugate (SPC). Avermectin (AVM) was hydrophobically encapsulated as a model drug to obtain SPC@AVM. The reaction between SPI and CMC was confirmed by infrared spectroscopy, thermal analysis and SDS-PAGE electrophoresis. The results of scanning electron microscopy showed that the average particle size of the drug-loaded microspheres was 129 nm and the shape of microspheres changed from block to spherical after the addition of AVM. After encapsulation of AVM, the absolute value of zeta potential was greater than 15 mV, which indicated better stability. Compared to AVM solution, SPC@AVM showed more wettability on the leaf surface and the contact angle on the leaves decreased from 71.64° to 57.33°. The maximum liquid holding capacity increased by 41.41%, from 8.85 to 12.52 mg cm$^{-2}$, which effectively reduced leaf loss. SPC@AVM also prevented UV photolysis, wherein the half-life was extended from 18 to 68 min when exposed to UV light. Moreover, toxicity tests showed that the encapsulation of AVM was beneficial to retain the insecticidal effect of AVM in the presence of ultraviolet light. The release rate of AVM showed pH responsiveness and the release rate under neutral conditions was faster than acidic and alkaline conditions. Moreover, the process conformed to the Weibull model.

## 1. Introduction

Pesticides are commonly used in agriculture to control diseases, pests and weeds from affecting crops [1]. Annual worldwide

usage of pesticides is in millions of tons, without which 40% of crop would be lost [2]. However, due to factors such as foliar loss, evaporation, photodegradation, etc., the amount of the active ingredient of the pesticide actually acting on the target is less than 10% [3–5]. This implies that the amount of harmful substances used is very high, which has a negative impact on the environment and hence is a matter of great concern [6,7].

Scientists have proposed two different approaches to solve this problem. The first one is to replace highly toxic organochlorine pesticides with bio-pesticides, which are biodegradable [8]. Some steps have been taken towards achieving this, for example, to obtain avermectin (AVM) by fermentation of fungi to prevent aphids or to kill pests and extraction of azadirachtin from plants for use as insecticides [9,10]. The second approach is to improve the utilization of pesticides by processing, mainly to reduce leaf loss, degradation and control their release [11,12]. Zhou & Zhang [13] prepared glucosylamide-based tetrasiloxane surfactant and added it to an AVM emulsion, which reduced the contact angle of the AVM emulsion on the spinach leaves and increased its diffusibility. Chen et al. [14] grafted styrene, methyl methacrylate and butyl acrylate onto sodium carboxymethyl cellulose (CMC) to prepare grafted polymer nanoparticles for sustained release of AVM. The anti-ultraviolet photolysis ability of AVM was improved and the release time of AVM was also prolonged.

The nano-material as a carrier of active ingredient possesses better permeability, thermal stability and biodegradability compared to conventional pesticide formulation. It improves the solubility of the active ingredient and facilitates its intelligent release [15,16]. Compared to synthetic polymeric materials, bio-based polymers are often preferred due to non-toxicity, high swelling in water, biodegradability, biocompatibility and ecofriendly properties [17,18]. Soy protein is an abundant by-product of soya bean oil industry [19]. In aqueous solution, soy protein possesses a hydrophilic shell and a hydrophobic core. Heating causes structural unfolding and denaturation of proteins, thereby increasing their surface hydrophobicity [20]. Teng et al. [21] synthesized soy protein nanoparticles by desolvation method for loading curcumin and the encapsulation efficiency achieved was up to 97%. However, the process required a large amount of ethanol as solvent, and toxic glutaraldehyde was used as a cross-linking agent to improve the stability of protein nanoparticles.

Polysaccharide–protein complexes and coacervates are biopolymeric systems that have been used over the past decades for encapsulation of numerous active ingredients [22]. In our previous work, [23] the carrier of feather keratin-hyaluronic acid (FK-HA) was prepared by Maillard reaction for encapsulation of AVM. The material showed high surface charge and was capable of homogeneous dispersion. FK-HA encapsulated AVM in a better manner, effectively slowed down the photolysis rate of AVM, and released the drug in response to pH.

In this work, soy protein isolate (SPI) and CMC were used as raw materials and $N'$-ethylcarbodiimide hydrochloride/$N$-hydroxysuccinimide (EDC/NHS) was used as the catalyst to form a stable amide bond between them, after which AVM was encapsulated as a model drug. CMC was obtained by modification of cellulose, which is the most abundant resource in nature. The molecular chain contained a large number of hydroxyl groups and so it was highly viscous in aqueous solution [24]. Using CMC as a pesticide carrier, a large number of hydroxyl groups were introduced, which promoted the formation of hydrogen bonds between the carrier and polar groups on the surface of the leaves, thereby improving the diffusibility at the leaf surface [25]. The CMC containing a large number of carboxyl groups was linked to the surface of the SPI. SPI was denatured by thermal deformation process and it underwent rearrangement due to hydrophobicity. Pores were formed due to static electricity, which permitted the entry and exit of AVM molecules. Encapsulation could effectively protect AVM from UV radiation, slow down its photolysis rate, reduce the contact angle of the solution on the leaf surface and increase its diffusibility. As SPI is an ampholyte, the release of the drug can also be controlled by pH, which improves the utilization of AVM.

# 2. Experimental set-up

## 2.1. Materials

SPI was obtained by extraction of de-fatted soy protein powder, provided by Tianli Food Co. Ltd (Anyang, China), according to previously reported methods [26]. CMC (USP grade) with a viscosity of 300–800 mPa s, sodium dodecyl sulfate (SDS) (electrophoresis grade), $N$-(3-Dimethylaminopropyl)-$N'$-ethylcarbodiimide hydrochloride (EDC) and the $N$-hydroxysuccinimide (NHS) were purchased from Aladdin Reagent Co. Ltd (Shanghai, China). The molecular weight of CMC, measured by gel

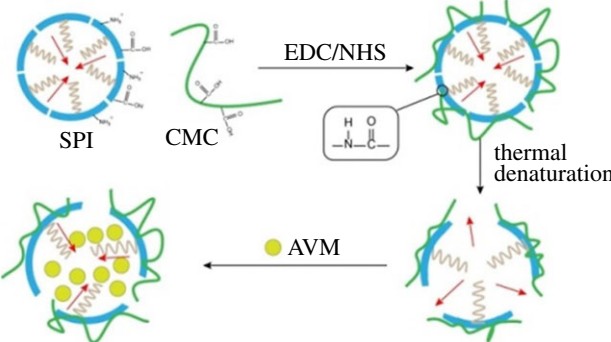

**Figure 1.** Mechanism of SPC synthesis.

**Figure 2.** Route for the synthesis of SPC@AVM.

permeation chromatography, was 216 kDa. AVM (96%) was provided by Hebei Weiyuan Biochemical Co. Ltd (Hebei, China). Hydrochloric acid, sodium hydroxide, absolute ethanol, methanol, potassium bromide and other chemicals were purchased from Tianjin Damao Chemical Reagent Co. Ltd (Tianjin, China), all of which were of analytical grade. Acrylamide (30%), Coomassie Brilliant Blue fast staining solution, Rainbow 245 plus Spectral standard protein (Maker) were purchased from Solarbio Technology Co. Ltd (Beijing, China) Tris–HCl buffer (1.5 M, pH 8.8), Tris–HCl buffer (1 M, pH 6.8) and sample buffer were purchased from Biosharp Biotechnology Co. Ltd (Hefei, China).

## 2.2. Preparation of SPC@AVM

EDC (0.25 g) and NHS (0.125 g) were weighed accurately and dissolved in deionized water to obtain 25 ml EDC/NHS solution. SPI (0.5 g) was dissolved in 50 ml deionized water and filtered to remove insoluble particles. Then CMC, with a certain mass ratio with respect to SPI, was added and dissolved by stirring. The pH of the above mixture was adjusted to 5.5 using dilute hydrochloric acid, followed by addition of 5 ml of EDC/NHS solution. The reaction was stirred for 24 h. The reaction solution was then transferred to a dialysis bag with a molecular weight cut-off of 5000 Da for 24 h to remove EDC and NHS. To obtain a pure SPC, the dialysed solution was placed in a refrigerator at −40°C for 24 h and then freeze-dried. The mechanism of SPC synthesis is shown in figure 1.

AVM (0.25 g) was dissolved in 25 ml of absolute ethanol. SPC was dissolved in 50 ml of deionized water to obtain a solution with SPC concentration of 1 mg ml$^{-1}$, the pH was adjusted to 7 and it was heated to 80°C for 1 h. When the temperature of the solution reached room temperature, 2 ml of AVM ethanol solution was added to it and stirred for 0.5 h to obtain SPC@AVM solution. The route for the synthesis of SPC@AVM was shown in figure 2.

## 2.3. Characterization

Graft ratio (GR): after completion of the reaction between SPI and CMC, the pH of the solution was adjusted to 4.5 (isoelectric point of SPI) by hydrochloric acid, and the suspension was centrifuged at 12 000 r.p.m. for 10 min, the supernatant was removed, freeze-dried for 24 h. The GR was calculated according to the formula (2.1). As shown in table 1, in SPC-11, SPC-41 and SPC-81, the graft ratios of CMC on SPI were 148.12%, 36.12% and 19.23%, respectively.

$$GR = \frac{m_2}{m_1 - m_3},$$ (2.1)

**Table 1.** Graft ratio of CMC on SPI with various $m_{SPI} : m_{CMC}$ ratios.

| sample | $m_{SPI} : m_{CMC}$ | graft ratio/% |
|---|---|---|
| SPC-11 | 1 : 1 | 148.12 ± 0.71 |
| SPC-41 | 1 : 0.25 | 36.12 ± 0.11 |
| SPC-81 | 1 : 0.125 | 19.23 ± 0.32 |

where $m_2$ is the feeding amount of CMC, $m_1$ is the feeding amount of SPI and $m_3$ is the mass of SPI not reacting with CMC.

Molecular weight ($M_w$) of a protein can be determined based on the extent of migration of a protein, after the complexation with a strong cationic detergent SDS separated on SDS-polyacrylamide gel electrophoresis (SDS-PAGE) [27]. Sample buffer (20 µl) was added to SPC solution (80 µl) and heated in boiling water bath for 10 min and then centrifuged at 12 000 r.p.m. for 1 min. The resulting SDS-PAGE gel contained 15% separating gel and 5% stacking gel. Then, the sample was added into a gel separation well and electrophoresis was conducted using an electrophoresis apparatus (DYCZ-24DN, Beijing Liuyi Inc., China). It was then stained with Coomassie Brilliant Blue dye. After dyeing, the gel was decolorized by adding deionized water.

Fourier-transform infrared (FTIR) spectra of the samples were obtained on a Spectrum-100 instrument (PerkinElmer Inc., USA), using KBr pellet method, in the spectral range of 4000–450 cm$^{-1}$. The samples for scanning electron microscopy (SEM) were prepared in the following manner: the sample solution was diluted 10 times, dropped on a substrate and then sprayed with gold, and observed under a scanning electron microscope (SU8010, Hitachi, Japan). Thermal stabilities of the materials were determined by thermogravimetric analysis (TGA) and differential scanning calorimetry (DSC), using TGA2 (Mettler Toledo, Switzerland) from 40°C to 600°C at a heating rate of 10°C min$^{-1}$ under a nitrogen atmosphere (20 ml min$^{-1}$), using Q200 differential scanning calorimeter (TA Co., USA) from 40°C to 180°C at a heating rate of 10°C min$^{-1}$ under a nitrogen atmosphere (20 ml min$^{-1}$). Zeta potentials and particle sizes of SPC@AVM were measured using laser particle size analyser (90-plus, Brookhaven, USA) after 10-fold dilution of the samples. To determine the contact angles of the samples, SPC@AVM solution was diluted to an AVM concentration of 100 mg l$^{-1}$, and the same concentration of AVM solution was used as control. About 5–10 µl of the sample was dropped onto the leaf surface, and the contact angle was measured using a contact angle meter (Theta, Biolin Scientific Co. Ltd, Sverige).

## 2.4. Encapsulation efficiency

According to a previous method [28], SPC@AVM solution was centrifuged at 12 000 r.p.m. for 10 min, and 1 ml of the supernatant was diluted to 25 ml with absolute ethanol. The absorbance was measured by an ultraviolet spectrophotometer (Lambda 365, PerkinElmer Instrument Co. Ltd, USA) at 245 nm wavelength. The concentration of AVM was calculated from the standard curve using the formula $A = 0.03319C - 0.00591$ ($R^2 = 0.999$). The amount of free AVM was calculated, and the entrapment efficiency $EE$ of AVM was calculated using the formula

$$EE = \frac{m_{\text{total AVM}} - m_{\text{free AVM}}}{m_{\text{total AVM}}},$$ (2.2)

where $m_{\text{total AVM}}$ is the total mass of AVM, and $m_{\text{free AVM}}$ is the mass of AVM after centrifugation and filtration.

## 2.5. Liquid holding capacity of foliage

Fresh leaves of vegetables were washed with deionized water and then cut into pieces of $2 \times 2$ cm dimensions for testing. The evenly sized leaves were soaked in the sample solution for 5 s and then lifted vertically using tweezers. They were weighed after the liquid stopped dripping. The liquid holding capacity (LHC) per unit area was calculated using the formula

$$LHC = \frac{M_1 - M_0}{2A},$$ (2.3)

where $M_0$ and $M_1$ represent the weights of the leaves before and after soaking, respectively, and $A$ represents the area of the leaves.

## 2.6. Protection against UV light

The SPC@AVM and 10 mg ml$^{-1}$ AVM ethanol solutions were diluted to 100 mg l$^{-1}$ with deionized water. The source for photolysis of AVM was a photochemical reactor (Shanghai Dusi Instrument Co. Ltd, China). The sample solution (50 ml) was placed under a 300 W mercury lamp ($E_{max}$ = 365 nm) at a distance of 15 cm from the source. At specific intervals, 1 ml of the sample was transferred to a brown volumetric flask, diluted to 25 ml with absolute ethanol and its absorbance was measured at 245 nm. The concentration of AVM in the sample at different time intervals was determined from the standard curve $A = 0.03319C - 0.00591$ ($R^2 = 0.999$). The remaining ratio (RR) of AVM in the sample was calculated using the formula

$$\text{RR} = \frac{c_i}{c_o},\tag{2.4}$$

where $c_0$ is the initial concentration of AVM, and $c_i$ is the concentration of AVM at different time points.

## 2.7. Sustained release of AVM

SPC@AVM solution (5 ml) was transferred into a dialysis bag and placed in a conical flask. Then, 50% ethanol–water solution (50 ml) was added at 36°C. At different intervals of time ($t$), 1 ml aliquot of the solution was transferred to a 25 ml volumetric flask. An equal volume of 40% aqueous ethanol solution was then added to the conical flask. The sample aliquot taken in the volumetric flask was diluted with 40% aqueous ethanol to the mark. The absorbance of the diluted sample solution was measured by an ultraviolet spectrophotometer at 245 nm. The AVM concentration of sample solution was calculated from the standard curve $A = 0.0299C + 0.00527$ ($R^2 = 0.999$), and the cumulative release rate was calculated using formula (2.5) [29].

$$R_i = \begin{cases} c_i \times \dfrac{0.05}{m_{AVM}} \, (i = 1) \\ c_i \times \dfrac{0.05}{m_{AVM}} + \displaystyle\sum_{i=1}^{i-1} c_i \times \dfrac{0.001}{m_{AVM}} \, (i > 1) \end{cases},\tag{2.5}$$

where $c_i$ is mass concentration (mg l$^{-1}$) of AVM for each sample at different time intervals, $m_{AVM}$ is the total mass of AVM in a conical flask.

Two factors, difference factor ($f_1$) and the similarity factor ($f_2$), were proposed to compare the differences of release curves [30]. $f_1$ represents the per cent difference between the two curves at every point of time. It measures the relative error between the two curves, while $f_2$ is a logarithmic reciprocal square root transformation of the sum of squared error and is a measure of similarity in per cent dissolution between the curves. $f_1$ and $f_2$ were obtained from formulae (2.6) and (2.7) [31].

$$f_1 = 100 \left[ \frac{\left( \sum_{t=1}^{n} |R_i - R_0| \right)}{\sum_{t=1}^{n} R_i} \right],\tag{2.6}$$

$$f_2 = 50 \log \left\{ \left[ 1 + \left( \frac{1}{n} \right) \sum_{t=1}^{n} (R_i + R_0)^2 \right]^{-0.5} \times 100 \right\},\tag{2.7}$$

where $n$ is the number of intervals in the experimentation, $R_0$ is the release rate from the reference group at specific time intervals and $R_i$ is the release rate from the control group at specific time intervals.

## 2.8. Toxicity tests

The solutions obtained above, SPC@AVM and AVM, were diluted with deionized water to 200 mg l$^{-1}$ concentration. They were both exposed to 300 W using a mercury lamp ($E_{max}$ = 365 nm) for 15 min to obtain corresponding UV/SPC@AVM and UV/AVM solutions. Using them, solutions were prepared which had AVM concentrations of 200, 100, 50, 25, 12.5 and 6.25 mg l$^{-1}$ and insecticidal activities of these formulations were tested. The leaves were cut into homogeneous pieces and dipped into the

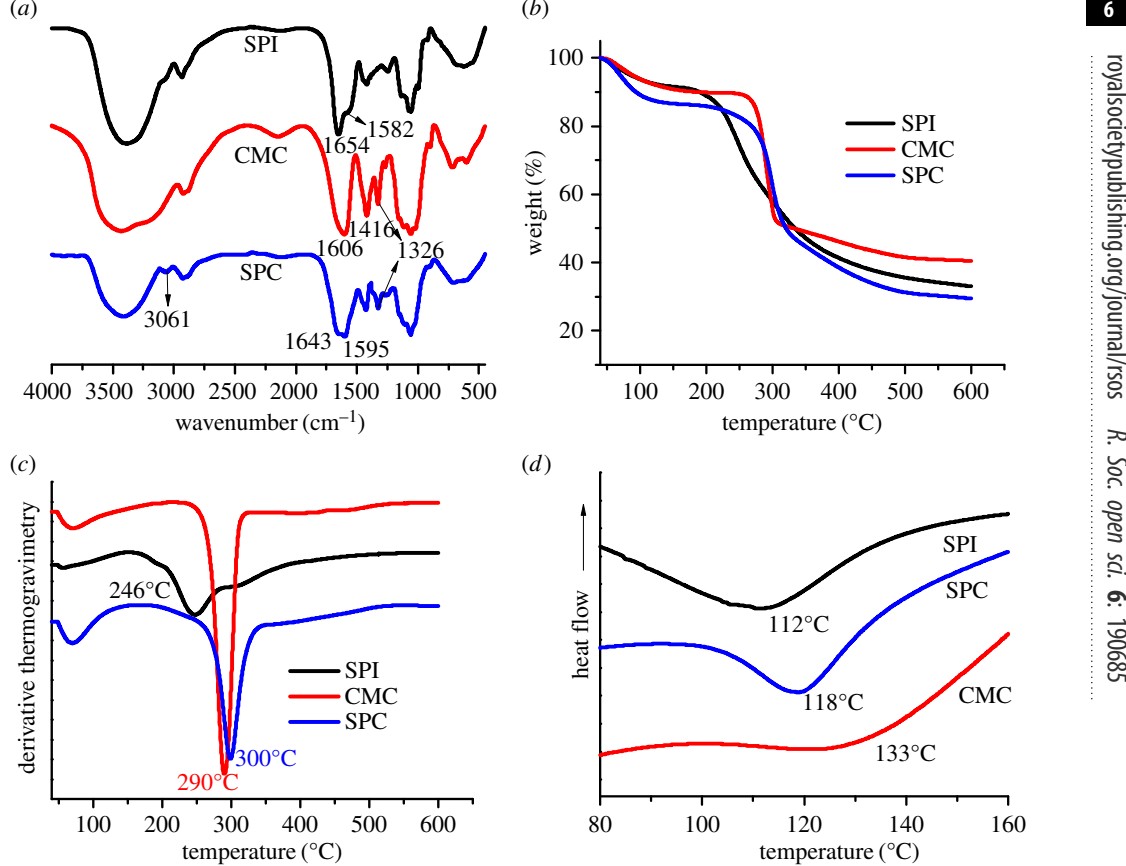

**Figure 3.** (a) FTIR spectra of SPI, CMC and SPC. (b,c) TG and DTG plots of SPI, CMC and SPC. (d) DSC plots of SPI, CMC and SPC.

sample solution for 5 min before they were removed and dried naturally. Then, the dried leaves along with 10 *Plutella xylostella* moths were placed in a Petri dish. These untreated leaves served as the blank experimental groups. The survival rate of *plutella xylostella* moths was evaluated after 48 h. Mortality and corrected mortality were calculated according to the Abbott formula, and median lethal concentration ($LC_{50}$) and toxicity regression equations were calculated based on the probabilistic analysis.

## 2.9. Statistics

All measurements were performed in triplicates. The results obtained were presented as means ± s.e. Data were analysed by analysis of variance ($p < 0.05$) using the SPSS software package.

# 3. Results and discussion

## 3.1. FTIR analysis

Figure 3a shows the FTIR spectra of SPI, CMC and SPC. The broad peak at 3415 cm$^{-1}$ in the spectrum of SPI was due to the N–H stretching vibrations of amide groups along with hydrogen bonding [32]. Peaks at 1654 cm$^{-1}$ and 1582 cm$^{-1}$ were assigned to the stretching vibrations of peptide bonds in SPI. The absorption peaks at 1606 and 1416 cm$^{-1}$ in the CMC were attributed to asymmetric and symmetric stretching vibrations of carboxylate salt, respectively [33]. In the case of SPC, the absorption peaks of amide I and amide II appeared at 1645 cm$^{-1}$ and 1595 cm$^{-1}$, respectively. These peaks were wider than those in the spectrum of SPI and were also shifted; the broadening of these peaks was due to the formation of additional amide bonds in addition to the existing peptide bonds. A new absorption peak at 3061 cm$^{-1}$ was characteristic of the amide bond. Both these findings indicated the successful formation of the amide bond between SPI and CMC [34]. In addition to these, in the spectrum of SPC, peak due to in-plane bending vibration of the hydroxyl group of CMC appeared at 1326 cm$^{-1}$, suggesting that a reaction occurred between SPI and CMC.

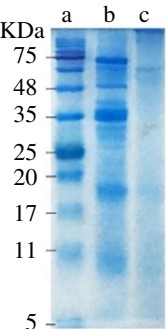

**Figure 4.** Results of SDS-PAGE gel electrophoresis of the Maker (a), SPI (b) and SPC (c).

## 3.2. Thermal analysis

The mass loss curve during heating of SPI, CMC and SPC is shown in figure 3b. Weight loss at around 100°C in all samples could be attributed to the evaporation of residual water. CMC is a polymer formed from the same monomer, and thermal decomposition occurs mainly at 260–300°C, which is a narrow range. The thermal decomposition of SPI is mainly concentrated in the range of 200–400°C, because the protein composition is extremely complex. For SPC, significant mass loss due to water evaporation can be observed around 100°C, which is due to the increase in the number of hydrophilic groups and the enhancement of water retention capacity after SPI was connected to CMC. The decomposition of SPC was mainly concentrated in 250–400°C. In order to further study the observed weight loss, the derivative thermogravimetry (DTG) of SPI, CMC and SPC was studied, and the results are shown in figure 3c. The weight loss peaks at 246°C and 290°C correspond to decomposition temperatures of SPI and CMC, respectively. However, compared to SPI and CMC, the decomposition temperature of SPC was increased. During degradation, initially, non-covalent bonds, such as intermolecular and intramolecular hydrogen bonds, electrostatic bonds and bonds due to hydrophobic interactions were broken. Subsequently, covalent bonds between C–N, C(O)-NH and C(O)-NH$_2$ were broken as temperature increased [35,36]. SPC had significantly more covalent bonds in its molecular chain than SPI and CMC, and the decomposition temperature of SPC was higher than CMC and SPI.

Variation in glass transition temperature is an effective indicator of compatibility between SPI and CMC. Figure 3d shows the DSC plots of CMC, SPI and SPC. The glass transition temperatures of SPI and CMC were 112°C and 133°C, respectively. SPC showed a single glass transition temperature at 118°C, indicating that SPC was more stable than SPI. A single glass transition in the DSC heating curves indicated that SPI and CMC were highly compatible [37].

## 3.3. SDS-PAGE analysis

Figure 4 shows the SDS-PAGE analyses of standard $M_W$ markers (line a), SPI (line b) and SPC (line c). In the case of line b, bands corresponding to molecular weights of 75, 48, 35, 18 and 10 kDa were observed, which was consistent with previous reports by Conti *et al.* [38] However, in case of line c, several bands of SPI had disappeared, and a new molecular weight band appeared at the top. This band due to increased molecular weight indicated that CMC and SPI had successfully reacted to form SPC [39].

## 3.4. Morphological analysis

Figure 5a–c shows the morphologies of SPC before and after heating and of synthesized SPC@AVM ($m_{SPI}$ : $m_{CMC}$ = 1 : 0.25). Figure 5d–f, respectively, represents the particle size distributions of SPC before and after heating and SPC@AVM, which were determined using Image—Pro Plus 6.0 software. The SPC particles were spherical in shape, had an average particle diameter of 136 nm and were connected to each other. This is due to the fact that SPI is a globular protein with an outer hydrophilic group and an internal hydrophobic group [40]. CMC is a long-chain polymer that reacts with several globular proteins and creates a link between particles. After heating, the SPC appeared crystalline with an average particle diameter of 125 nm. The disulfide bonds within the protein absorbed energy and broke, which resulted in the unfolding of the polypeptide chain and recombination of different fragments hydrophobically [41]. In the recombination process, the formation of crystal shape and decrease in particle size were mainly due to the good film-forming property of CMC [42].

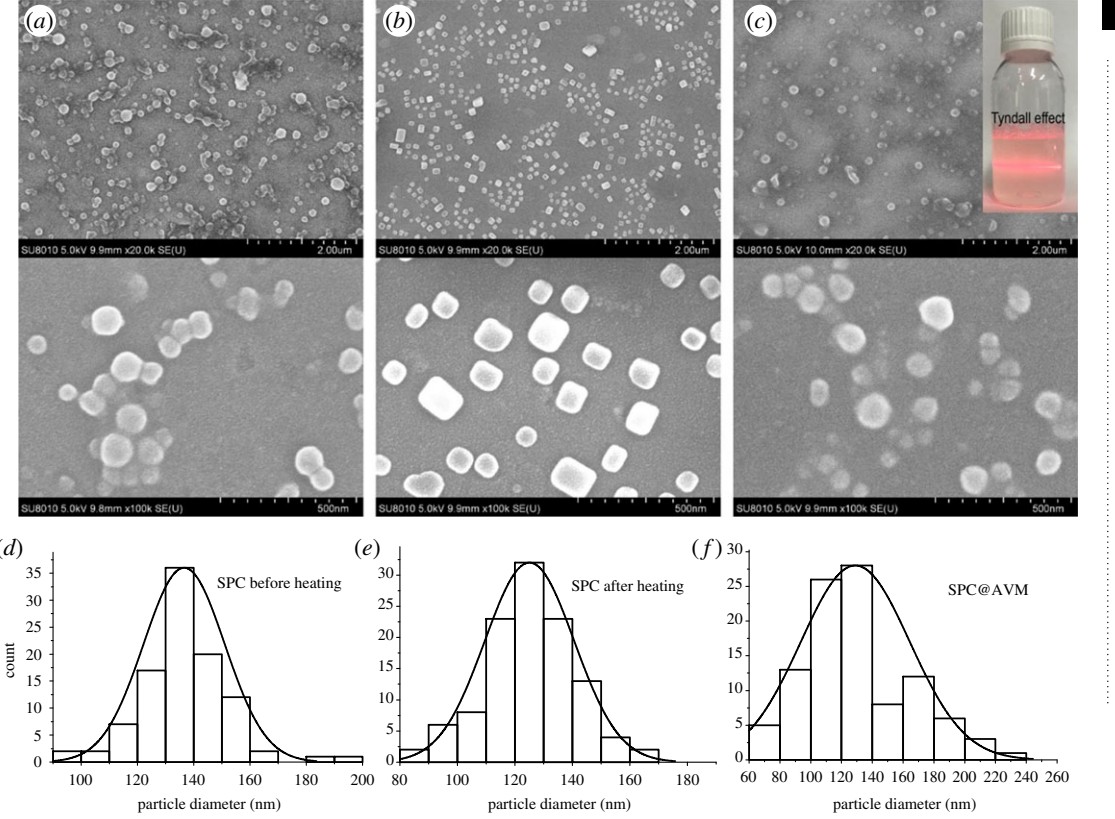

**Figure 5.** (*a*–*c*) SEM images of SPC before and after heating and SPC@AVM. (*d*–*f*) Particle size distributions of SPC before and after heating and SPC@AVM.

**Table 2.** Encapsulation efficiency, particle size and zeta potential of SPC@AVM under different conditions.

| sample | pH | zeta potential (mV) | average size (nm) | *EE* (%) |
|---|---|---|---|---|
| SPC-81@AVM | 7 | −20.71 ± 0.18 | 155.68 ± 3.43 | 34.8 ± 0.05 |
| SPC-41@AVM | 7 | −24.87 ± 1.75 | 166.91 ± 0.56 | 40.8 ± 0.22 |
| SPC-11@AVM | 7 | −43.37 ± 0.33 | 233.65 ± 3.91 | 44.3 ± 0.13 |
| SPC-11@AVM | 5 | −6.97 ± 0.21 | 264.88 ± 10.87 | 44.9 ± 0.06 |
| SPC-11@AVM | 9 | −53.30 ± 2.04 | 220.40 ± 0.55 | 33.5 ± 0.07 |

As shown in figure 5*c*, the entry of AVM into SPC was followed by the transformation of the particles into spherical shape. The SPC@AVM solution showed good dispersibility, and a significant Tyndall effect under laser irradiation, indicating that SPC@AVM formed micelles in aqueous solution. In addition, comparison of figure 5*e,f* showed that the addition of AVM to SPC resulted in an increase in the average particle diameter, from 125 to 129 nm. This is because the hydrophobic group of SPC itself formed a cavity in the hydrophobic action. When AVM entered this hydrophobic cavity, the original hydrophobic balance was broken and a new hydrophobic cavity was formed due to enhanced hydrophobic interactions. The reformed hydrophobic core had a greater attraction for the SPC segment, which exceeded the tendency of CMC to form a film. Therefore, the SPC was re-transformed into a spherical shape from a block shape.

## 3.5. Size, zeta potential, encapsulation efficiency of the composites

### 3.5.1. Effect of CMC concentration

Table 2 shows differences in zeta potentials, particle sizes and encapsulation efficiencies of the composites prepared using different CMC concentrations. Increase in CMC content had a great

influence, which included an increase in absolute charge of the system, increase in encapsulation efficiency and an increase in particle size of AVM. An increase in absolute zeta potential value with an increase in CMC content was as expected normal, due to a large number of negatively charged carboxylate ions in the CMC molecular chain, which improved solution stability. The hydrodynamic sizes of SPC-41@AVM microspheres were a little larger than the corresponding diameters of microspheres from SEM characterization. This indicated that the SPC@AVM microspheres were swollen in water, due to the hydrophilic nature of microspheres [43]. The reason for the increase in particle size is that the site of the reaction of SPI with CMC increases, with the amount of CMC carrying a large amount of carboxyl groups, and the increase in the number of carboxyl groups allowed for more CMC molecules to be attached to SPI to form larger particles. Moreover, the increase in the encapsulation efficiency of AVM also contributed to the increase in particle size. Owing to the surface charge repulsions, the recombination of the microspheres under hydrophobic conditions became more difficult. This resulted in an increased spacing between the fragments of SPC and consequently an increase in the particle size of SPC microspheres [44]. This facilitated the easy entry of AVM. Therefore, the higher the CMC content, the higher was its packaging efficiency.

### 3.5.2. Effect of pH

The zeta potential value of the system varies significantly with pH. The particle size of SPC-11@AVM increased with an increase in pH. At pH 5, the repulsive forces between the particles were small, and agglomeration caused an increase in particle size. With an increase in pH, the degree of protonation of the amino group decreased, whereas the degree of deprotonation of carboxyl group increased, accompanied by an increase in negative charge of the system. Therefore, the zeta potential of SPC-11@AVM was minimal at pH 9. As repulsive forces between particles increased, the dispersibility of the particles increased, and the system was more stable. Under acidic conditions, the number of positively charged amino groups was greater and there was an electrostatic force of attraction between them and the negatively charged AVM. Hence, encapsulation efficiency was significantly improved in acidic conditions as compared to the alkaline conditions. As mentioned above, the increase in repulsive forces between different SPC fragments led to larger gaps between SPC segments. This made the entry of AVM into the interiors of the microspheres easier. However, an increase in potential also caused an increase in the electrostatic repulsion between the surfaces of the microsphere and AVM. Therefore, the encapsulation efficiency between pH 5 and pH 7 did not change significantly. As the pH was increased to 9, the electrostatic repulsions between the microspheres and AVM increased again, due to which the encapsulation rate decreased.

## 3.6. Foliage wettability analysis

The wettability of the sample solution on the plant surface was studied by measuring the contact angle of the sample solution on the foliage. As shown in figure 6a, the contact angle made by AVM solution on the foliage was significantly reduced as compared to the deionized aqueous solution. This was because ethanol used to dissolve AVM was compatible with the hydrophobic portion of foliage. The contact angles of SPC-11@AVM, SPC-41@AVM and SPC-81@AVM solutions were 56.45°, 62.70° and 68.15° on the leaves. They were all less than that of deionized water (92.47°) and AVM solution (72.19°), which indicates the better wettability performance after encapsulation in SPC. Further, with an increase in CMC concentration, the contact angle decreased, indicating that CMC had an affinity towards leaf surface. This was because the main leaf component is a polysaccharide, containing many hydroxyl groups. Moreover, the leaf surface is covered with a waxy substance, its main components being fatty acids, fatty alcohols and fatty aldehydes [45]. These polar functional groups on the surface of the leaves interact with functional groups of CMC.

As shown in figure 6b, the maximum LHC of SPC-11@AVM was significantly more than the AVM aqueous solution. There was an increase of 41.41%, from 8.85 to 12.52 mg cm$^{-2}$. This variation of LHC with CMC concentration was consistent with a change in contact angle. When the leaves were pulled out from the sample solution vertically, the solution that remained on the hydrophobic leaf surface gathered into droplets and rolled on the leaves. The increase in contact angle increased the distance of the centre of gravity of the droplet from the leaf surface and reduced the interactive forces between the leaves and droplets.

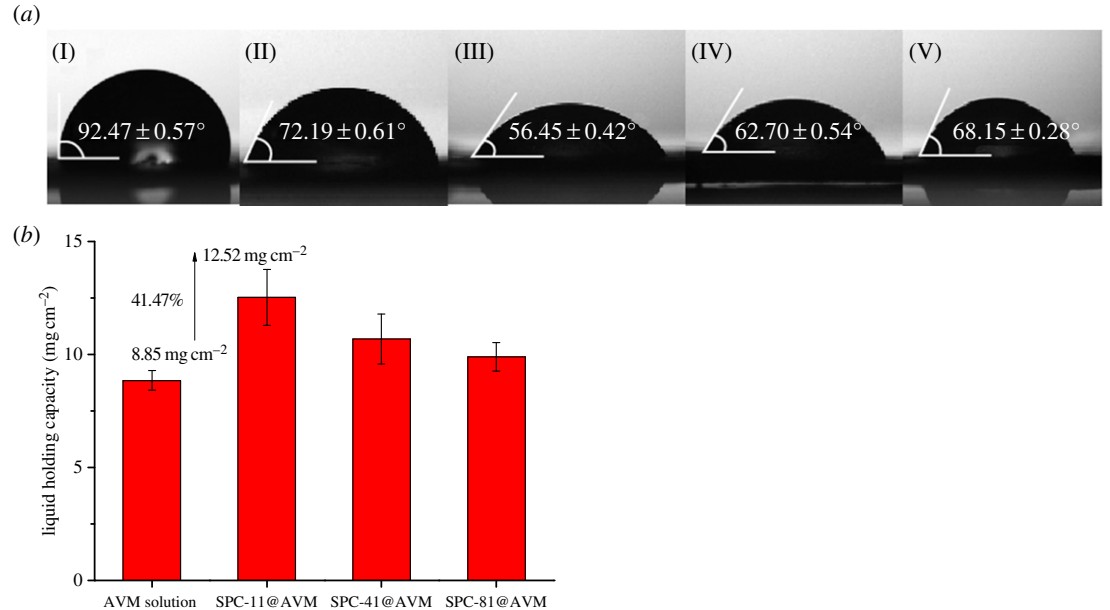

**Figure 6.** (*a*) Contact angles of deionized water (I), AVM solution (II), SPC-11@AVM (III), SPC-41@AVM (IV) and SPC-81@AVM (V) on the leaf surface. (*b*) Liquid holding capacities of the sample solutions on the leaf surface.

## 3.7. Performance analysis of sustained release

### 3.7.1. Effect of CMC concentration

Figure 7*a,b* shows the AVM release curves of different SPC@AVM samples and the AVM release curves of SPC@AVM samples under different pH conditions, respectively. Initially, the release of AVM is exponential, because the unencapsulated AVM in the solution passes quickly through the dialysis bag under osmotic pressure, which would be beneficial for immediate insecticidal effects. Subsequently, the rate gradually slowed down, which would be useful for prolonging the time of insecticidal action and reducing the required amount of pesticides.

Table 3 shows the results of statistical analysis using similar factors and dissimilar factors based on the release curve of SPC-11@AVM under different pH conditions. The release factors of SPC-41@AVM and SPC-81@AVM were similar, and the values of the statistical analysis results were similar. $f_1$ less than 15 and $f_2$ greater than 50 indicate that the difference was not significant compared to that of SPC-11@AVM, and the release rate was only slightly higher than them [46,47].

Unlike the encapsulation process of drugs, when AVM molecule enters the interiors of the microspheres, in addition to constraint by hydrophobic forces, the negative charge of carboxymethyl group on microsphere surface also generates electrostatic repulsion, making it more difficult to escape. As described above, SPC-11@AVM carried a greater number of negative charges and its release rate was slower. SPC-41@AVM and SPC-81@AVM had less access to carboxymethyl groups and carried a lesser number of negative charges, and so their release rates were faster. In addition, particle size was also closely related to the release rate of AVM. A large particle size indicated that the AVM needed to travel a longer distance during the release process and the release rate was slower.

### 3.7.2. Effect of pH

As shown in figure 7*b*, the release rate of SPC@AVM at pH 7 was faster than the release rates under acidic and alkaline conditions. Statistical analysis showed that under acidic conditions, $f_1$ less than 15, $f_2$ greater than 50, indicating that they were not significantly different from the release rates under neutral conditions. SPC@AVM had a large particle size under acidic conditions and AVM required a longer time to pass from the inside to the outside, which resulted in the slow release rate of AVM. In addition to this, the positive charge on SPC under acidic conditions increased and its mutual attraction with the negatively charged AVM increased, which was one of the main reasons for the slower escape rate of AVM. Under neutral conditions, diameters of the particles were significantly reduced relative to those under acidic conditions. The AVM could pass through the particles in a

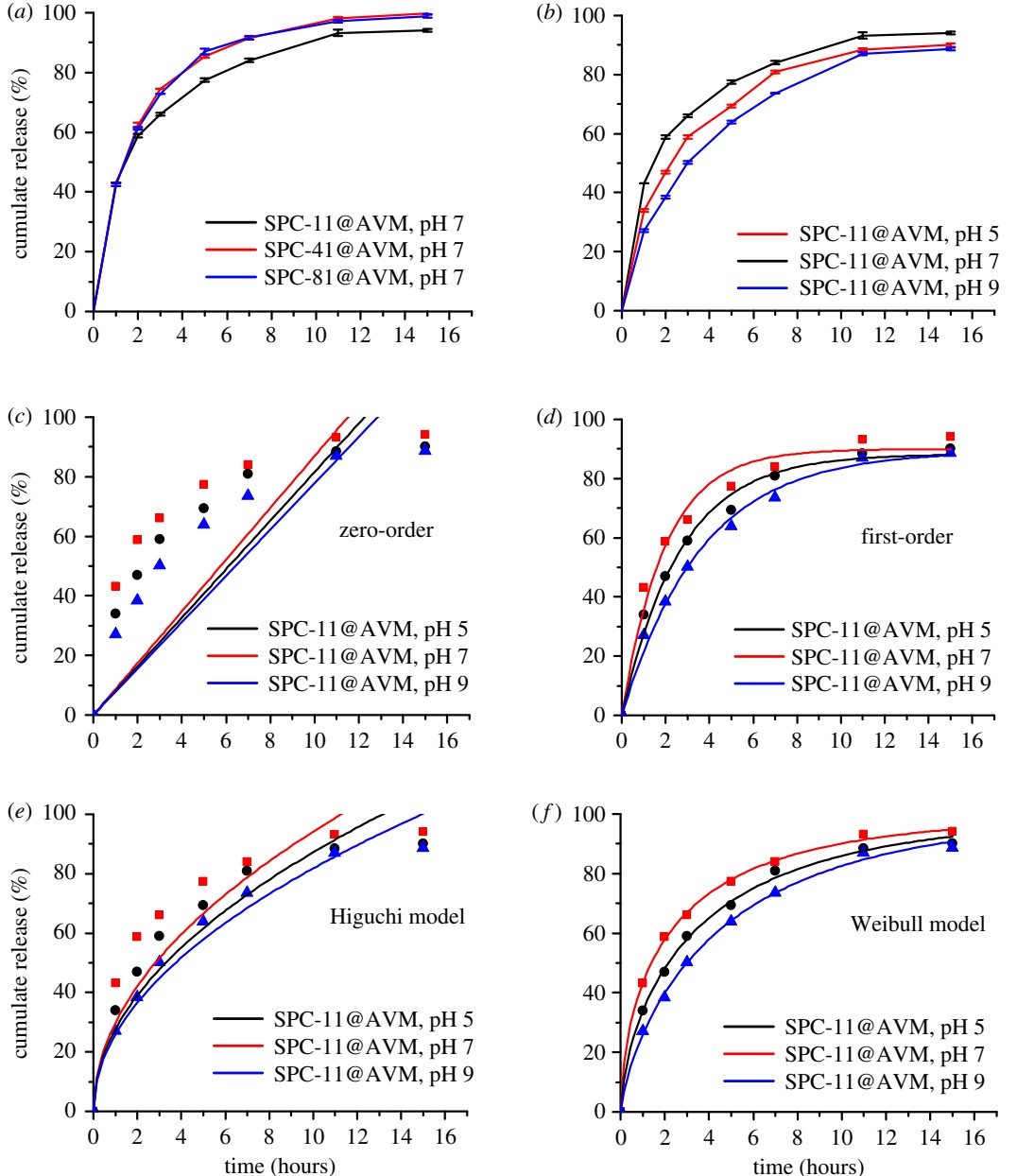

**Figure 7.** (*a*) Sustained release properties of different SPC@AVM composites. (*b*) Sustained release properties of SPC@AVM composites at different pH values. (*c–f*) Model fitting of the data for drug release at different pH values.

**Table 3.** Results of statistical analysis of $f_1$ and $f_2$.

| reference sample | control sample | $f_1$ | $f_2$ |
|---|---|---|---|
| SPC-11@AVM, pH = 7 | SPC-41@AVM, pH 7 | 6.98 | 60.47 |
| | SPC-81@AVM, pH 7 | 6.54 | 61.22 |
| | SPC-11@AVM, pH 5 | 10.35 | 56.04 |
| | SPC-11@AVM, pH 9 | 20.52 | 43.31 |

shorter period of time, and the release rate was accelerated. Under alkaline conditions, $f_1$ greater than 15 indicated a significant change from its release under neutral conditions. SPC@AVM had a smaller particle size under alkaline conditions than under acidic conditions, and hence AVM had a shorter transit time. However, the negative charge on SPC surface increased and the negative charge on the microsphere

**Table 4.** Fitting results for sustained release of AVM at different pH values.

| fitting model | formula | pH | $a_0$ | $a_1$ | $R^2$ |
|---|---|---|---|---|---|
| zero-order | $y = a_0 x$ | 5 | 0.082 | — | 0.825 |
| | | 7 | 0.087 | — | 0.787 |
| | | 9 | 0.078 | — | 0.869 |
| first-order | $y = a_0(1 - \exp(-a_1 x))$ | 5 | 0.883 | 0.374 | 0.986 |
| | | 7 | 0.899 | 0.508 | 0.972 |
| | | 9 | 0.893 | 0.275 | 0.991 |
| Higuchi | $y = a_0 x^{0.5}$ | 5 | 0.276 | — | 0.904 |
| | | 7 | 0.298 | — | 0.813 |
| | | 9 | 0.259 | — | 0.965 |
| Weibull | $y = 1 - \exp(-(x_0^a)/a_1)$ | 5 | 0.679 | 2.433 | 0.997 |
| | | 7 | 0.612 | 1.768 | 0.998 |
| | | 9 | 0.766 | 3.343 | 0.998 |

surface increased the electrostatic repulsion of the internal AVM, making it more difficult to escape, resulting in a slower release rate.

### 3.7.3. Study of release kinetics

The release mechanisms of AVM under different pH conditions was further investigated and the corresponding data parameters were fitted to zero-order and first-order equations, Higuchi kinetic model, [48] and Weibull model. The zero-order model describes a drug delivery system, independent of the initial concentration [49]. The first-order model implies that the release of drug is proportional to its concentration [50]. If the release process conforms to the Huiguchi model, the release mechanism is considered to be a pure diffusion process of the drug from a carrier without erosion or swelling of the carrier [51]. The Weibull function emerges due to the creation of a concentration gradient near the releasing boundaries of the Euclidian matrix or due to the 'fractal kinetics' behaviour associated with the fractal geometry of the environment [52,53].

As shown in table 4 and figure 7c–f, the AVM release curves followed the Weibull model, wherein the $R^2$ was greater than 0.99. When $0.35 < a_0 < 0.69$, the diffusion in the fractal or disordered matrix was different from the permeation group. When the $a_0$ value was greater than 0.69 and less than 0.75, the release followed the Fick diffusion model in the Euclidean or fractal space. When $0.75 < a_0 < 1$, along with diffusion in the normal Euclidean matrix, there is another release mechanism [54]. At pH 5 and pH 7, the $a_0$ was more than 0.35 and less than 0.69, indicating that the release of AVM was different from that of the percolation cluster. The release of AVM was not a movement in continuous media but has a process from the inside out, which is subject to many obstacles. At pH 9, $a_0$ was greater than 0.75, which indicated that the release of AVM was affected not only by Fick diffusion but also by another release mechanism. In addition to concentration, the other effect was electrostatic interaction, which had a capturing effect on AVM.

## 3.8. Anti-UV property

It is well known that AVM is susceptible to decomposition particularly under ultraviolet light. Therefore, improving the stability of AVM is the key to improve the utility of pesticides. Figure 8 shows that the photolysis of AVM solution under UV irradiation was significantly faster than that of AVM after SPC encapsulation, with a half-life of only 17 min. With an increase in CMC content, the photolysis rate was faster, and the half-lives of SPC-11@AVM, SPC-41@AVM and SPC-81@AVM were 30, 50 and 68 min, respectively. Considering these results, SPC@AVM was believed to slow down the process of photolysis of AVM under UV irradiation. There are two reasons for this: one was that SPC prevented the ultraviolet light from directly illuminating AVM, [55] and the other was because CMC and SPI

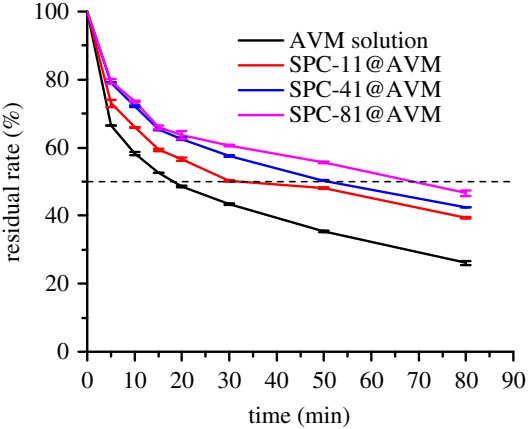

**Figure 8.** Photolysis of AVM solution and AVM encapsulated composites with time.

**Table 5.** Results of toxicity test.

| sample | toxicity regression eq. | $LC_{50}$ (mg l$^{-1}$) | 95% confidence limit | $R^2$ |
|---|---|---|---|---|
| AVM | $y = 1.2722x + 3.5760$ | 13.16 | 5.86–29.55 | 0.9671 |
| UV/AVM | $y = 1.2289x + 3.1356$ | 32.90 | 21.06–51.40 | 0.9722 |
| SPC@AVM | $y = 0.9475x + 3.9083$ | 14.20 | 5.49–36.74 | 0.9470 |
| UV/SPC@AVM | $y = 1.4691x + 3.0748$ | 20.44 | 12.44–33.59 | 0.9765 |

contained many chromophores (carbonyl and carboxyl groups), [56] which absorbed ultraviolet light and reduced the light intensity.

## 3.9. Toxicity analysis

The toxicities of AVM and UV/AVM were compared. As shown in table 5, the toxicity regression equations for AVM and UV/AVM were $y = 1.2722x + 3.5760$ and $y = 1.2289x + 3.1356$, and their $LC_{50}$ values were 13.16 and 32.90 mg l$^{-1}$, which indicated a significant difference in toxicity. The decline in AVM toxicity was due to photodegradation of AVM, which was the main reason for the low utilization of AVM during its use.

In previous reports, the $LC_{50}$ value of AVM, which was processed into a slow-release formulation was usually higher than that of the original drug due to its slower release rate [57]. As shown in table 5, compared with AVM, the virulence of SPC@AVM has decreased a little, but not significantly, which indicated that the encapsulation of AVM does not affect the virulence of pure AVM. After UV irradiation, AVM decomposes and the virulence decreases, the $LC_{50}$ of UV/SPC@AVM was significantly higher compared with SPC@AVM.

The $LC_{50}$ value of UV/SPC@AVM was higher than that of AVM and also much smaller than UV/AVM, which indicated that the toxicity of SPC@AVM after illumination was intermediate between them. From the above anti-UV experiment, it can be confirmed that the encapsulation of AVM could effectively slow down its photo-degradation rate. Therefore, after irradiation, the amount of free AVM in SPC@AVM was greater than that in the untreated AVM, and consequently the insecticidal effect was good.

## 4. Conclusion

In this study, CMC was grafted onto the SPI surface to obtain SPC, which was then heated to increase its hydrophobicity. The reaction between SPI and CMC was demonstrated by different characterization methods. SPC@AVM could improve the residual amount of AVM on the leaf surface, reduce leaf loss and provide AVM with excellent UV resistance, which could effectively extend the half-life of AVM. Toxicity tests confirmed that encapsulation of AVM could effectively reduce the attenuation of AVM

toxicity under UV irradiation. During the release process, SPC@AVM showed pH-responsive property, and the release of AVM was consistent with the Weibull model under different pH conditions. This study aimed to construct a water-based drug-loading system for AVM, which is an environmentally unstable and insoluble pesticide. SPC@AVM could reduce the loss and degradation of pesticide and control the release of pesticides, which could effectively improve the utilization of pesticides by reducing the amount of pesticide required.

Data accessibility. The datasets supporting this article have been uploaded as the electronic supplementary material.

Authors' contributions. L.C. and H.Z. carried out the laboratory work, participated in data analysis, participated in the design of the study and drafted the manuscript; L.H. carried out the statistical analyses; H.C. collected field data; X.Z. conceived of the study, designed the study, coordinated the study and helped draft the manuscript. All authors gave final approval for publication.

Competing interests. We have no competing interests.

Funding. This research was funded by National Natural Science Foundation of China (grant no. 21576303), Natural Science Foundation of Guangdong Province (grant nos. 2016A030313375, 2017A030311003), Science and Technology Program of Guangzhou, China (grant no. 201903010011).

Acknowledgements. No one contributed to the study but did not meet the authorship criteria.

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
