## [Reviewer comments · Royal Society Open Science]

Review History

RSOS-190685.R0 (Original submission)

Review form: Reviewer 1

Is the manuscript scientifically sound in its present form?

Yes

Are the interpretations and conclusions justified by the results?

Yes

Is the language acceptable?

Yes

Is it clear how to access all supporting data?

Yes

Do you have any ethical concerns with this paper?

No

Have you any concerns about statistical analyses in this paper?

No

Recommendation?

Accept with minor revision (please list in comments)

Comments to the Author(s)

This paper investigated the synthesis and characterization of drug-loaded microspheres from carboxymethyl cellulose and soy protein isolate. It is found that the AVM loaded microspheres demonstrated more wettability on the leaf surface than AVM solution. Furthermore, SPC@AVM also prevented UV photolysis when exposed to UV light. The experiments were carried out using established methods, and the results support the conclusions. Publication after minor revision is suggested.

1. The authors used FTIR analysis of prove the successful preparation of SPC. Obviously, this is a weak point. FTIR only provide a quantitative analysis for the samples. The yield and polymer structure of SPC were not provided from this characterization. More characterizations are suggested to be carried out and more discussions are added in the manuscript.
2. The authors claimed that the initial decomposition temperatures of SPI, CMC, and SPC were 160 ° C, 188 ° C, and 232 ° C, respectively. Please check these conclusions and figure 3b. It seems that SPC demonstrate a low thermal stability than other two samples. By the way, how the authors define the initial decomposition temperatures?
3. The sustained release of different SPC@AVM composites in Figure 7 seems to show slightly difference. But the error bars for each data point are suggested to be added in the Figure 7. The same revisions also should be made in Figure 8.
4. The author claimed that a single glass transition in the DSC heating curves indicated that SPI and CMC were highly compatible. These conclusions are questionable. Each curve in Figure 3 (d) shows a wide and broad peak. Are you sure this is only a single peak? If this is the truth, please provide the purity of SPC.

Review form: Reviewer 2

Is the manuscript scientifically sound in its present form?

No

Are the interpretations and conclusions justified by the results?

Yes

Is the language acceptable?

Yes

Is it clear how to access all supporting data?

Yes

Do you have any ethical concerns with this paper?

No

Have you any concerns about statistical analyses in this paper?

No

Recommendation?

Major revision is needed (please make suggestions in comments)

Comments to the Author(s)

This paper describes a natural nano-composites by self-assembly with soy protein isolate (SPI) and carboxymethyl cellulose (CMC) for efficiently loading, protecting and releasing pesticide AVM. Briefly, the manuscript had a clear thought, the preparation process and result analysis are detailed, and the project shows great potentials in agricultural production. However, there were still some questions needs to be improved and detailed comments are the following:

1. In this paper, CMC was grafted onto SPI, thus, the graft ratio needs to be given.
2. The author using the UV absorption to measure AVM content, and the standard curve seems to be the same in '2.5 Encapsulation efficiency' and '2.6 Protection against UV light'. Please carefully check.
3. Page 8 line 5, what means '...CMC is a long-chain polymer...'
4. In Fig. 5d-5f, one transverse captions should be used.
5. In Fig. 7a, the pH value of release is missing; In Fig. 7b-f, please give the specific type of SPC@AVM sample.
6. In toxicity analysis, the LC50 value of UV/SPC@AVM is higher than that of AVM solution, however, the reason is not clear. It may be degradation loss by UV irradiation or the slow-release from composites. The SPC@AVM group should be used to illustrate this point.

Decision letter (RSOS-190685.R0)

28-May-2019

Dear Sir Chen:

Title: Soy protein isolate-carboxymethyl cellulose conjugates with pH sensitivity for sustained avermectin release

Manuscript ID: RSOS-190685

The editor assigned to your manuscript has now received comments from reviewers. We would like you to revise your paper in accordance with the referee and Subject Editor suggestions which can be found below (not including confidential reports to the Editor). Please note this decision does not guarantee eventual acceptance.

Please submit your revised paper before 20-Jun-2019. Please note that the revision deadline will expire at 00.00am on this date. If we do not hear from you within this time then it will be assumed that the paper has been withdrawn. In exceptional circumstances, extensions may be possible if agreed with the Editorial Office in advance. We do not allow multiple rounds of revision so we urge you to make every effort to fully address all of the comments at this stage. If deemed necessary by the Editors, your manuscript will be sent back to one or more of the original reviewers for assessment. If the original reviewers are not available we may invite new reviewers.

To revise your manuscript, log into <http://mc.manuscriptcentral.com/rsos> and enter your Author Centre, where you will find your manuscript title listed under "Manuscripts with Decisions." Under "Actions," click on "Create a Revision." Your manuscript number has been

appended to denote a revision. Revise your manuscript and upload a new version through your Author Centre.

RSC Associate Editor:
Comments to the Author:
(There are no comments.)

RSC Subject Editor:
Comments to the Author:
(There are no comments.)

Reviewers' Comments to Author:
Reviewer: 1

Comments to the Author(s)

This paper investigated the synthesis and characterization of drug-loaded microspheres from carboxymethyl cellulose and soy protein isolate. It is found that the AVM loaded microspheres demonstrated more wettability on the leaf surface than AVM solution. Furthermore, SPC@AVM also prevented UV photolysis when exposed to UV light. The experiments were carried out using established methods, and the results support the conclusions. Publication after minor revision is suggested.

1. The authors used FTIR analysis of prove the successful preparation of SPC. Obviously, this is a weak point. FTIR only provide a quantitative analysis for the samples. The yield and polymer structure of SPC were not provided from this characterization. More characterizations are suggested to be carried out and more discussions are added in the manuscript.

2. The authors claimed that the initial decomposition temperatures of SPI, CMC, and SPC were 160 ° C, 188 ° C, and 232 ° C, respectively. Please check these conclusions and figure 3b. It seems that SPC demonstrate a low thermal stability than other two samples. By the way, how the authors define the initial decomposition temperatures?
3. The sustained release of different SPC@AVM composites in Figure 7 seems to show slightly difference. But the error bars for each data point are suggested to be added in the Figure 7. The same revisions also should be made in Figure 8.
4. The author claimed that a single glass transition in the DSC heating curves indicated that SPI and CMC were highly compatible. These conclusions are questionable. Each curve in Figure 3 (d) shows a wide and broad peak. Are you sure this is only a single peak? If this is the truth, please provide the purity of SPC.

Reviewer: 2

Comments to the Author(s)

This paper describes a natural nano-composites by self-assembly with soy protein isolate (SPI) and carboxymethyl cellulose (CMC) for efficiently loading, protecting and releasing pesticide AVM. Briefly, the manuscript had a clear thought, the preparation process and result analysis are detailed, and the project shows great potentials in agricultural production. However, there were still some questions needs to be improved and detailed comments are the following:

1. In this paper, CMC was grafted onto SPI, thus, the graft ratio needs to be given.
2. The author using the UV absorption to measure AVM content, and the standard curve seems to be the same in '2.5 Encapsulation efficiency' and '2.6 Protection against UV light'. Please carefully check.
3. Page 8 line 5, what means '...CMC is a lo4ng-chain polymer...'
4. In Fig. 5d-5f, one transverse captions should be used.
5. In Fig. 7a, the pH value of release is missing; In Fig. 7b-f, please give the specific type of SPC@AVM sample.
6. In toxicity analysis, the LC50 value of UV/SPC@AVM is higher than that of AVM solution, however, the reason is not clear. It may be degradation loss by UV irradiation or the slow-release from composites. The SPC@AVM group should be used to illustrate this point.

Author's Response to Decision Letter for (RSOS-190685.R0)

See Appendix A.

Decision letter (RSOS-190685.R1)

24-Jun-2019

Dear Sir Chen:

Title: Soy protein isolate-carboxymethyl cellulose conjugates with pH sensitivity for sustained avermectin release

Manuscript ID: RSOS-190685.R1

It is a pleasure to accept your manuscript in its current form for publication in Royal Society Open Science. The chemistry content of Royal Society Open Science is published in collaboration with the Royal Society of Chemistry.

RSC Associate Editor
Comments to the Author:
(There are no comments.)

Reviewer(s)' Comments to Author:

Appendix A

Dear. Editors and Reviewers:

We appreciate the time and detailed suggestions provided by each reviewer and have incorporated the suggested changes into the manuscript to the best of our ability. We have responded specifically to each suggestion below. All changes made to the manuscript (Manuscript Number: RSOS-190685) are marked in the revised manuscript.

Reviewers' comments and our answers

Comments of Reviewer #1:

This paper investigated the synthesis and characterization of drug-loaded microspheres from carboxymethyl cellulose and soy protein isolate. It is found that the AVM loaded microspheres demonstrated more wettability on the leaf surface than AVM solution. Furthermore, SPC@AVM also prevented UV photolysis when exposed to UV light. The experiments were carried out using established methods, and the results support the conclusions. Publication after minor revision is suggested.

1. The authors used FTIR analysis of prove the successful preparation of SPC. Obviously, this is a weak point. FTIR only provide a quantitative analysis for the samples. The yield and polymer structure of SPC were not provided from this characterization. More characterizations are suggested to be carried out and more discussions are added in the manuscript.

Answer: Thanks for reviewer's advice. We have made a quantitative analysis of the grafting ratio of CMC on SPI. After completion of the reaction between SPI and CMC, the pH of the solution was adjusted to 4.5 (isoelectric point of SPI) by hydrochloric acid, and the suspension was centrifuged at 12000 rpm for 10 minutes, the supernatant was removed, freeze-dried for 24 hours. The graft ratio (*GR*) was calculated according to the formula (1). AS shown in table 1, in SPC-11, SPC-41, and SPC-81, the grafting ratio of CMC on SPI were 148.12%, 36.12%, and 19.23%, respectively. Besides, we utilized SDS-PAGE analysis to compare the molecular weight of SPC and also confirmed the formation of SPC.

$$GR = m_2 / (m_1 - m_3) \quad (1)$$

Where, m_2 is the feeding amount of CMC, m_1 is the feeding amount of SPI, and m_3 is the mass of SPI not reacting with CMC.

Table 1 Graft ratio of CMC on SPI with various $m_{SPI} : m_{CMC}$ ratio

Sample	$m_{SPI} : m_{CMC}$	Graft ratio/%
SPC-11	1:1	148.12±0.71
SPC-41	1:0.25	36.12±0.11
SPC-81	1:0.125	19.23±0.32

2. The authors claimed that the initial decomposition temperatures of SPI, CMC, and SPC were 160 ° C, 188 ° C, and 232 ° C, respectively. Please check these conclusions and figure 3b. It seems that SPC demonstrate a low thermal stability than other two samples. By the way, how the authors define the initial decomposition temperatures?

Answer: Thanks for reviewer's question. In fact, the thermal stability of SPC is not weaker than that of SPI. The mass loss of SPC in the early stage is mainly due to the moisture in the sample. When CMC is combined with SPI, the hydrophilic groups are significantly increased, and the water retention capacity of SPC is strong, so the residual moisture of the sample become more

obvious. For the initial decomposition temperature, it is the temperature at which the material begins to decompose obtained from tangent of TG curve. However, when the decomposition temperature of the material is around 100 ° C, the initial decomposition temperature is often difficult to determine due to the influence of moisture in the sample. Therefore, we removed the analysis of the initial decomposition temperature in the manuscript and re-analyzed the thermal stability of the material from the main decomposition temperature range.

3. The sustained release of different SPC@AVM composites in Figure 7 seems to show slightly difference. But the error bars for each data point are suggested to be added in the Figure 7. The same revisions also should be made in Figure 8.

Answer: Thanks for reviewer's advice. We have added error bars in the Figure 7a, 7b and Figure 8. However, Figs 7c-7f were not required error bars due to model fitting data.

4. The author claimed that a single glass transition in the DSC heating curves indicated that SPI and CMC were highly compatible. These conclusions are questionable. Each curve in Figure 3 (d) shows a wide and broad peak. Are you sure this is only a single peak? If this is the truth, please provide the purity of SPC.

Answer: Thanks for reviewer's question. We re-tested the DSC using different instruments and found that SPI, CMC, and SPC also showed single glass transitions which were still wide and broad peaks. According to the literature,^[1-3] the glass transition curves of some polymers (such as proteins and polysaccharides) were a wide and broad peak due to the complex structure of natural polymers. In addition, we found that the purity of SPC in the test sample was 83.76% according to formula (2).

$$Purity = (m_1 - m_2) / m_1 \quad (2)$$

Where, m_1 is the total mass of sample, m_2 is the mass of SPI not reacting with the CMC, which can be obtained by the isoelectric point deposition method.

References:

[1] Arik Kibar, E. A. & Us, F. 2013 Thermal, mechanical and water adsorption properties of corn starch - carboxymethylcellulose/methylcellulose biodegradable films. *Journal of Food Engineering* 114, 123-131.

[2] Dai, H., Ou, S., Liu, Z. & Huang, H. 2017 Pineapple peel carboxymethyl cellulose/polyvinyl alcohol/mesoporous silica SBA-15 hydrogel composites for papain immobilization. *Carbohydrate Polymers* 169, 504-514.

[3] Su, J., Huang, Z., Yuan, X., Wang, X. & Li, M. 2010 Structure and properties of carboxymethyl cellulose/soy protein isolate blend edible films crosslinked by Maillard reactions. *Carbohydrate Polymers* 79, 145-153.

Comments of Reviewer #2:

This paper describes a natural nano-composites by self-assembly with soy protein isolate (SPI) and carboxymethyl cellulose (CMC) for efficiently loading, protecting and releasing pesticide AVM. Briefly, the manuscript had a clear thought, the preparation process and result analysis are detailed, and the project shows great potentials in agricultural production. However, there were still some questions needs to be improved and detailed comments are the following.

1. In this paper, CMC was grafted onto SPI, thus, the graft ratio needs to be given.

Answer: Thanks for reviewer’s advice. We have made a quantitative analysis of the grafting ratio of CMC on SPI. After completion of the reaction between SPI and CMC, the pH was adjusted to 4.5 (isoelectric point of SPI) with hydrochloric acid, and the suspension was centrifuged at 12000 rpm for 10 minutes, the supernatant was removed, freeze-dried for 24 hours. The graft ratio (GR) was calculated according to the formula (1). AS shown in table 1, in SPC11, SPC41, and SPC81, the grafting ratio of CMC on SPI were 148.12%, 36.12%, and 19.23%, respectively.

$$GR = m_2 / (m_1 - m_3) \quad (1)$$

Where, m_2 is the feeding amount of CMC, m_1 is the feeding amount of SPI, and m_3 is the mass of SPI not reacting with CMC.

Table 1 Graft ratio of SPC with various $m_{SPI} : m_{CMC}$ ratio

Sample	$m_{SPI} : m_{CMC}$	Graft ratio/%
SPC-11	1:1	148.12±0.71
SPC-41	1:0.25	36.12±0.11
SPC-81	1:0.125	19.23±0.32

2. The author using the UV absorption to measure AVM content, and the standard curve seems to be the same in ‘2.5 Encapsulation efficiency’ and ‘2.6 Protection against UV light’. Please carefully check.

Answer: Thanks for reviewer’s question. We dilute the samples using absolute ethanol in the “2.5 Encapsulation efficiency” and “2.6 Protection against UV light”, so the standard curve was consistent.

3. Page 8 line 5, what means ‘...CMC is a lo4ng-chain polymer...’

Answer: Thanks for reviewer’s question. We have changed the “CMC is a lo4ng-chain polymer” to “CMC is a long-chain polymer”.

4. In Fig. 5d-5f, one transverse captions should be used.

Answer: Thanks for reviewer’s advice. We have used transverse captions in Figs. 5d-5f. As shown in Figs. 5d-5f.

Figs. 5d-5f Particle size distributions of SPC before and after heating and SPC@AVM

5. In Fig. 7a, the pH value of release is missing; In Fig. 7b-f, please give the specific type of SPC@AVM sample.

Answer: Thanks for reviewer’s advice. We added the pH value of release in Figure 7a; in the text and Fig. 7b-f we added the type of SPC@AVM sample used in the sustainable release

performance test. As shown in Figs. 7a-7f.

Fig. 7a Sustained release properties of different SPC@AVM composites

Fig. 7b Sustained release properties of SPC@AVM composites at different pH values

Figs. 7c-7f Model fitting of the data for drug release at different pH values

6. In toxicity analysis, the LC50 value of UV/SPC@AVM is higher than that of AVM solution; however, the reason is not clear. It may be degradation loss by UV irradiation or the slow-release from composites. The SPC@AVM group should be used to illustrate this point.

Answer: Thanks for reviewer’s advice. We performed the experiment again and supplemented the virulence analysis of SPC@AVM. The results show in Table 5. It showed that the virulence difference between SPC@AVM and AVM was not significant. SPC@AVM maintains high insecticidal activity after illumination due to the encapsulation of AVM by SPC. It has been revised in the manuscript.

Table 5 Results of toxicity test

Sample	Toxicity regression eq.	LC_{50} /mg/L	95% confidence limit	R^2
AVM	$y = 1.2722x + 3.5760$	13.16	5.86-29.55	0.9671
UV/AVM	$y = 1.2289x + 3.1356$	32.90	21.06-51.40	0.9722
SPC@AVM	$y = 0.9475x + 3.9083$	14.20	5.49-36.74	0.9470
UV/SPC@AVM	$y = 1.4691x + 3.0748$	20.44	12.44-33.59	0.9765

Special thanks to you for your good comments. We appreciate for Editors/Reviewers' warm work earnestly, and hope that the correction will meet with approval.

With best regards,

Hongjun Zhou

Xinhua Zhou

School of Chemistry and Chemical Engineering, Zhongkai University of Agriculture and Engineering, Guangzhou 510225, China

E-mail: hongjunzhou@163.com

E-mail: cexinhuazhou@163.com